

# A retrospective study of prevalence and pattern of international consensus on ANA patterns among patients with hepatitis C virus infection

Shun-Wen Hsiao[1,*], Chuan-San Fan[1,*], Hsu-Heng Yen[1,2,3,4], Siou-Ping Huang[1], Yang-Yuan Chen[1,5,6] and Pei-Yuan Su[1]

[1] Division of Gastroenterology, Changhua Christian Hospital, Changhua, Taiwan
[2] General Education Center, Chienkuo Technology University, Changhua, Taiwan
[3] Department of Electrical Engineering, Chung Yuan Christian University, Taoyuan, Taiwan
[4] College of Medicine, National Chung Hsing University, Taichung, Taiwan
[5] Division of Gastroenterology, Yuanlin Christian Hospital, Changhua, Taiwan
[6] Department of Hospitality Management, MingDao University, Changhua, Taiwan
[*] These authors contributed equally to this work.

Corresponding author
Pei-Yuan Su, 111252@cch.org.tw

## ABSTRACT

**Background**. A previous study reported a 30% prevalence of various autoantibodies among patients with hepatitis C virus (HCV) infection. The International Consensus on Anti-Nuclear Antibody (ANA) Patterns was recently introduced to classify ANA patterns based on immunoassay on HEp-2 cells. There is no previous report with this newly developed classification to evaluate patients with HCV infection. The study aims to study the prevalence and pattern of ANA patterns among HCV-infected patients.

**Methods**. We retrospectively analyzed the medical records of patients with HCV infection from September 2020 to June 2021 at our institution. A positive ANA is defined as a titer of more than 1:320. We compared patient features among the positive and negative groups.

**Results**. Overall, 258 patients were enrolled—184 patients with negative ANA and 74 patients (28.7%) with positive ANA. The mean age was 67.3 in ANA positive group and 61.2 ANA negative group. Female was prominent with ANA positive and accounted for 63.5%. The most detected ANA pattern was AC-1(homogeneous) (25.9%), followed by AC-4(fine speckled) (25.2%) and AC-21(anti-mitochondrial antibody) (9.6%). In ANA positive group, we found a trend of lower HCV viral load (5.72 $\log_{10}$ IU/ML vs. 6.02 $\log_{10}$ IU/ML), lower alanine aminotransferase level (39.5 U/L vs. 44 U/L), and higher advanced fibrosis (F3 and F4) (38.5% vs. 26.1%). In addition, higher positive ANA (more than 1:640) is significantly associated with lower estimated glomerular filtration rate (eGFR) (77.76 vs. 87.94 mL/min/1.73 $m^2$, $P = 0.044$).

**Conclusions**. A high prevalence (28.7%) of ANA was found in patients with chronic hepatitis C. The presence of positive ANA is not related to the severity of their hepatic manifestation. However, higher positive ANA was significantly associated with lower eGFR.

## INTRODUCTION

Chronic hepatitis C (CHC) is one of the significant causes of cirrhosis and hepatocellular carcinoma (HCC) worldwide. It is estimated that 74% of patients with CHC had at least one extrahepatic clinical manifestation, such as lymphoma, cryoglobulinemia, glomerulonephritis, arthritis, autoimmune thyroiditis, type 2 diabetes mellitus, and Sjögren's syndrome (SjS) (*Giuggioli et al., 2017*; *Cacoub & Saadoun, 2021*; *Negro et al., 2015*; *Su et al., 2014*; *Vergani & Mieli-Vergani, 2013*). Researchers have found other autoantibodies in patients with CHC, including anti-antinuclear antibody (ANA), anti-smooth muscle antibody (ASMA), anti-mitochondrial antibody (AMA), and anti-liver/kidney microsomes type 1 (anti-LKM1). The prevalence of these autoantibodies was approximately 30% in patients with CHC (*Cassani et al., 1997*). ANA, detected by indirect immunofluorescence assay of HEp-2 cells (IIFA Hep-2), offers the titer of the fluorescence intensity and provides fluorescence patterns, which are helpful in diagnosing several systemic autoimmune diseases.

Two workshops are used for the standardization and classification of ANA patterns, namely, the International Autoantibody Standardization (IAS) and the International Consensus on ANA Patterns (ICAP) (*Agmon-Levin et al., 2014*; *Damoiseaux et al., 2015*). The IAS perspective promotes the standardization and improvement of autoantibodies in laboratory diagnosis. The ICAP classification presents four categories—negative, nuclear, cytoplasmic, and mitotic—and the 29 IIFA Hep-2 patterns with the clinical correlation (*Chan et al., 2022*; *Damoiseaux et al., 2019*). Each pattern is assigned a code below the descriptor, namely, anti-cell pattern 1 (AC-1) (*Chan et al., 2015*).

One decade ago, we used interferon (IFN)-based therapy to treat patients with hepatitis C virus (HCV) infection. Several researchers have reported that IFN-based therapies triggered the development of autoimmune diseases, extrahepatic manifestations, and autoantibodies (*Covini et al., 2012*; *Jadali, 2013*; *Kajiyama et al., 2012*; *Marazuela et al., 1996*; *Novembrino et al., 2014*). With the development of direct-acting antiviral agents (DAAs), the treatment of HCV infection has become feasible because DAAs showed a sustained virologic response rate >90%, less intolerance side effects, and short treatment duration compared with IFN-based therapies (*Chi et al., 2021*; *Yen et al., 2020*). Several case reports have shown a flare of autoimmune hepatitis or primary biliary cholangitis during or after DAA treatment (*Choi et al., 2020*; *Fianchi, Ponziani & Pompili, 2020*; *Matsumoto et al., 2018*; *Montón, Escudero & Pascual, 2020*; *Nguyen et al., 2018*). It is unknown whether the patient or the effect of DAAs triggers the autoimmunity. It is also important to know the autoimmunity status before treatment.

This study was designed to analyze the prevalence of ANA patterns in patients with CHC before DAA treatment and determine the association between different ANA patterns and patient characteristics.

## MATERIALS & METHODS

### Patient population

We retrospectively analyzed the medical records of patients who received anti-HCV therapy from September 2020 to June 2021 at our institution. The presence of ANA and the patient's HCV viremia level were checked before initiating anti-HCV treatment. Patients who underwent a test of blood ANA with a positive test for HCV viremia were included. In this study, 258 patients with viremia were enrolled. The body mass index was measured for direct-acting antiviral agent therapy at the first visit. The comorbidities were recorded according to the medical record.

### Method of ANA analysis

The ANA level was detected by the immunoassay of IIFA HEp-20-10 cells (EUROIMMUN, Luebeck, Germany) (*Voigt et al., 2012*). The serum samples were diluted with phosphate-buffered saline (PBS) starting from 1:10, 1:100, and 1:1000, and 30 µL of the diluted sample was drawn using a titer plate, overlaid onto a BIOCHIP slide, and incubated at room temperature for 30 min. The slides were washed with distilled water and soaked in PBS with Tween solution. Then, 25- µL anti-human globulin G labeled with fluorescein isothiocyanate was added, and the mixture was incubated at room temperature for 30 min. The slides were rewashed with distilled water and soaked in PBS with Tween solution for 5 min. All procedures were completed according to the manufacturers' protocol by experienced technicians. Trained technicians read the slides using a fluorescence microscope (IF Sprinter, EUROIMMUN). Two medical laboratory scientists identified the ANA patterns and titer to reduce the bias. The patients were deemed ANA-positive if the titer values were >1:320. The titer of 1:320 was deemed to be strong fluorescence intensity at 1:10 dilution, moderate fluorescence intensity at 1:100 dilution, but negative fluorescence intensity at 1:1000 dilution. Every fluorescence pattern with titer values >1: 320 was documented. The ANA patterns were differentiated according to the ICAP recommendations: Negative: AC-0; Nuclear pattern: AC-1 homogeneous, AC-2 dense fine speckled, AC-3 centromere, AC-4 fine speckled, AC-5 large/coarse speckled, AC-6 multiple nuclear dots, AC-7 few nuclear dots, AC-8 homogenous nucleolar, AC-9 clumpy nucleolar, AC-10 punctate nucleolar, AC-11 smooth nuclear envelop, AC-12 punctate nuclear envelope, AC-13 PCNA, AC-14 CENP-F, and AC-29 Topo-I; Cytoplasmic patterns: AC-15 fibrillar linear, AC-16 fibrillar filamentous, AC-17 fibrillar segmental, AC-18 discrete dots, AC-19 dense fine speckled, AC-20 fine speckled, AC-21 AMA, AC-22 Golgi, and AC-23 rods and rings; and mitotic patterns: AC-24 centrosome, AC-25 spindle fibers, AC-26 NuMA, AC-27 intercellular bridge, and AC-28 mitotic chromosomal (*Chan et al., 2022*).

### Liver stiffness and steatosis measurements

The liver stiffness measurements and controlled attenuation parameter (CAP) were performed using FibroScan (Echosens), which showed accuracy in assessing steatosis and fibrosis with an area under the receiver operating characteristic curve values ranging from 0.70 to 0.89 (*Eddowes et al., 2019*). The following liver stiffness cutoff values were used for staging: F0 <6.5 kPa; F1 <8 kPa; F2 <9.5 kPa; F3 <12 kPa, and F4 ≥ 12 kPa (*Wong et al.,*

*2010*). The diagnosis of non-alcoholic fatty liver disease was defined as the median value of CAP ≥ 248 dB/m (*Wong et al., 2010*).

## Ethical considerations

The Ethics Committee of Changhua Christian Hospital approved the study protocol used in this study (CCH IRB No: 220327), and the requirement for informed consent was waived because of the retrospective nature of this study.

## Statistical analysis

Data are expressed as numbers with percentages, medians with interquartile ranges (Q1–Q3) (The first quartile Q1: 25th percentile, the third quartile Q3: 75th percentile), or means ± standard deviations. The Kolmogorov–Smirnov test was used to determine whether data have a normal or non-normal distribution. Student's $t$-test or the Mann–Whitney U-test was used to compare continuous variables in the ANA(+) and ANA(-) groups. As appropriate, the frequencies of categorical variables were compared using the chi-square test or Fisher's exact test. Multivariate logistic regression analysis was performed to identify factors associated with ANA 1:640(+). Selected variables with $p$-values <0.10 from the crude mode (including age, sex (male), and estimated glomerular filtration rate (eGFR)) for the backward elimination procedure in the multivariate model. The results were considered statistically significant if the two-tailed $p$-value was <0.05 for all tests. Statistical analyses were performed using Statistical Package for the Social Sciences, version 22.0 (IBM Corp., Armonk, NY, USA).

## RESULTS

### Clinical features of the study population

Overall, 258 patients were enrolled for analysis. The characteristics of the patients are shown in Table 1. In the ANA(-) group, there were 184 patients, whereas, in the ANA(+) group, there were 74 patients. The mean age was 67.3 and 61.2 years in the ANA(+) and ANA(-) groups, respectively ($p = 0.001$). There was a female dominance in the ANA(+) group, accounting for 63.5% ($p = 0.023$). In the ANA(+) group, we found a lower HCV RNA viral load (5.72 $\log_{10}$ IU/ML *vs.* 6.02 $\log_{10}$ IU/ML; $p = 0.06$) and higher prevalence of advanced fibrosis (F3 and F4) (38.5% *vs.* 26.1%; $p = 0.098$).

### ANA pattern according to the ICAP

The most detected ANA pattern was AC-1 ($n = 35$), accounting for 47.3%, followed by AC-4 ($n = 34$), accounting for 45.9%; AC-21 ($n = 13$), accounting for 13/74; AC-8 ($n = 12$), accounting for 17.6%; and AC-27 ($n = 10$), accounting for 16.2% (Table 2). More than one ANA pattern was detected in 36 patients in Table 3. The most mixed ANA pattern was AC-1 and AC-4 ($n = 7$), followed by AC-1, AC-4, and AC-21 ($n = 3$) and AC-1, AC-4, and AC-27 ($n = 3$).

### The clinical correlation with high ANA levels in patients with chronic HCV infection

A high positive ANA level, with a titer value >1:640, was associated with older age, chronic kidney disease, and a significantly lower eGFR (77.76 *vs.* 87.94 mL/min/1.73 m$^2$; $p = 0.044$)

**Table 1  Characteristics of the patients with positive results of ANA.**

| Baseline characteristics | ANA 1:320 (+) | ANA(-) | *P*-value |
|---|---|---|---|
| Case number | 74 | 184 | |
| Age, years (Mean ± SD) | 67.3 ± 13.1 | 61.2 ± 12.9 | 0.001 |
| Gender male (*n*, %) | 27(36.5%) | 96(52.2%) | 0.023 |
| Body Mass Index (Mean ± SD) | 24.7 ± 3.7 | 24.7 ± 4.4 | 0.989 |
| Comorbidities | | | |
| Hypertension (*n*, %) | 34(45.9%) | 66(35.9%) | 0.133 |
| Diabetes mellitus (*n*, %) | 14(18.9%) | 37(20.1%) | 0.828 |
| Chronic kidney disease (*n*, %) | 6(8.1%) | 9(4.9%) | 0.378 |
| Cirrhosis | 14(18.9%) | 26(14.1%) | 0.336 |
| HCC (*n*, %) | 4(5.4%) | 5(2.7%) | 0.283 |
| Malignancy (*n*, %) | 8(10.8%) | 23(12.5%) | 0.706 |
| HCV RNA (log10 IU/ML) (Median(Q1-Q3)) | 5.72(4.44–6.39) | 6.02(5.01–6.52) | 0.060 |
| HCV Genotype (*n*, %) | | | 0.564 |
| 1 | 1(1.4%) | 7(3.8%) | 0.446 |
| 1a | 4(5.4%) | 15(8.2%) | 0.617 |
| 1b | 39(52.7%) | 72(39.1%) | 0.064 |
| 2(2,2a,2b) | 24(32.4%) | 65(35.3%) | 0.766 |
| 3 | 2(2.7%) | 6(3.3%) | 1.000 |
| 6 | 1(1.4%) | 7(3.8%) | 0.446 |
| Mixed | 0(0.0%) | 1(0.5%) | 1.000 |
| Unclassified | 3(4.1%) | 11(6.0%) | 0.763 |
| Baseline ALT (U/L) (Median(Q1-Q3)) | 39.5(22–82) | 44(27-91) | 0.574 |
| Baseline Bilirubin (mg/dL) (Median(Q1-Q3)) | 0.6(0.4–0.8) | 0.6(0.5–0.8) | 0.227 |
| Baseline eGFR (mL/min/1.73m2) (Median(Q1-Q3)) | 79.62(62.18–102.95) | 88.92(70.91–102.19) | 0.168 |
| Fibrosis (*n*, %) | | | 0.485 |
| F0 | 16(30.8%) | 58(43.3%) | 0.162 |
| F1 | 7(13.5%) | 18(13.4%) | 1.000 |
| F2 | 9(17.3%) | 23(17.2%) | 1.000 |
| F3 | 8(15.4%) | 13(9.7%) | 0.400 |
| F4 | 12(23.1%) | 22(16.4%) | 0.399 |
| F3 and F4 (*n*, %) | 20(38.5%) | 35(26.1%) | 0.098 |

(Table 4). Multivariable analysis (Table 5) also showed the significance of the relationship between high positive ANA levels and lower eGFR ($p = 0.008$).

# DISCUSSION

In our analysis, 28.7% of the patients with CHC were ANA-positive. According to the ICAP, the most detected ANA pattern was AC-1, followed by AC-4, AC-21, AC-8, and AC-27. Of the 74 ANA-positive patients, 36 had more than one ANA pattern. ANA-positive patients showed a lower HCV RNA viral load and higher prevalence of advanced fibrosis (F3 and

**Table 2** The distribution of ANA pattern in chronic HCV patients according to international consensus on ANA pattern.

| ANA pattern (≥ 1:320) | Number of positive/total positive samples ($N = 74$) | Percentage |
|---|---|---|
| Nuclear Homogeneous (AC1) | 35/74 | 47.3% |
| Nuclear dense fine speckled (AC2) | 0/74 | 0.0% |
| Centromere (AC3) | 4/74 | 5.4% |
| Nuclear fine speckled (AC4) | 34/74 | 45.9% |
| Nuclear large/coarse speckled (AC5) | 3/74 | 4.1% |
| Multiple nuclear dots (AC6) | 4/74 | 5.4% |
| Few nuclear dots (AC7) | 4/74 | 5.4% |
| Homogeneous nucleolar (AC8) | 12/74 | 16.2% |
| Clumpy nucleolar (AC9) | 2/74 | 2.7% |
| Punctate nucleolar (AC10) | 2/74 | 2.7% |
| Smooth nuclear envelope (AC11) | 1/74 | 1.4% |
| Punctate nuclear envelope (AC12) | 0/74 | 0.0% |
| PCNA-like (AC13) | 0/74 | 0.0% |
| CENP-F-like (AC14) | 0/74 | 0.0% |
| Cytoplasmic fibrilliar linear (AC15) | 0/74 | 0.0% |
| Cytoplasmic fibrilliar filamentous (AC16) | 2/74 | 2.7% |
| Cytoplasmic fibrillar segmental (AC17) | 0/74 | 0.0% |
| Cytoplasmic discrete dots/GW-body like (AC18) | 0/74 | 0.0% |
| Cytoplasmic dense fine speckled (AC19) | 0/74 | 0.0% |
| Cytoplasmic fine speckled (AC20) | 0/74 | 0.0% |
| Cytoplasmic reticular/AMA (AC21) | 13/74 | 17.6% |
| Polar/Golgi-like (AC22) | 0/74 | 0.0% |
| Rods and rings (AC23) | 2/74 | 2.7% |
| Centrosome (AC24) | 5/74 | 6.8% |
| Spindle fiber (AC25) | 0/74 | 0.0% |
| NuMA-like (AC26) | 2/74 | 2.7% |
| Intercellular bridge (AC27) | 10/74 | 13.5% |
| Mitotic chromosomal envelope (AC28) | 0/74 | 0.0% |
| Topo I-like (AC29) | 0/74 | 0.0% |

F4) without statistical significance. Higher positive ANA levels (>1:640) were significantly associated with a lower eGFR.

Patients with CHC are predisposed to have extrahepatic manifestations (*Vergani & Mieli-Vergani, 2013*). The most strongly associated disease is cryoglobulinemia (*Giuggioli et al., 2017*; *Comarmond, Cacoub & Saadoun, 2020*). Moreover, several systemic autoimmune diseases have also been reported, such as autoimmune thyroiditis (*Jadali, 2013*), glomerulonephritis (*Negro et al., 2015*), rheumatoid arthritis (*Su et al., 2014*), and SjS (*Wang et al., 2014*). The incidence of rheumatological manifestations in patients with CHC is estimated to be 38% (*Priora et al., 2021*). Besides, it has been noticed that several

**Table 3  Mixed ANA pattern in patients with chronic HCV patient.**

| ANA pattern (≥ 1:320) | Number/total positive samples (N = 74) | percentage |
|---|---|---|
| AC 1 & AC 4 | 7/74 | 9.5% |
| AC 1 & AC 4 & AC 21 | 3/74 | 4.1% |
| AC 1 & AC 4 & AC 27 | 3/74 | 4.1% |
| AC 1 & AC 4 & AC 24 | 2/74 | 2.7% |
| AC 1 & AC 6 | 1/74 | 1.4% |
| AC 1 & AC 8 | 1/74 | 1.4% |
| AC 1 & AC 21 | 1/74 | 1.4% |
| AC 4 & AC 5 | 1/74 | 1.4% |
| AC 4 & AC 10 | 1/74 | 1.4% |
| AC 4 & AC 24 | 1/74 | 1.4% |
| AC 4 & AC 27 | 1/74 | 1.4% |
| AC 5 & AC 27 | 1/74 | 1.4% |
| AC 7 & AC 21 | 1/74 | 1.4% |
| AC 8 & AC 9 | 1/74 | 1.4% |
| AC 8 & AC 11 | 1/74 | 1.4% |
| AC 26 & AC 27 | 1/74 | 1.4% |
| AC 1 & AC 4 & AC 5 | 1/74 | 1.4% |
| AC 4 & AC 8 & AC 21 | 1/74 | 1.4% |
| AC 4 & AC 21 & AC 27 | 1/74 | 1.4% |
| AC 1 & AC 4 & AC 7 & AC 27 | 1/74 | 1.4% |
| AC 1 & AC 4 & AC 8 & AC 23 | 1/74 | 1.4% |
| AC 1 & AC 4 & AC 10 & AC 21 | 1/74 | 1.4% |
| AC 1 & AC 4 & AC 24 & AC 27 | 1/74 | 1.4% |
| AC 4 & AC 6 & AC 7 & AC 8 | 1/74 | 1.4% |
| AC 1 & AC 4 & AC 6 & AC 7 & AC 8 & AC 21 | 1/74 | 1.4% |

autoimmune diseases may be triggered during interferon therapy for patients with CHC (*Pellicano et al., 2005*; *Wilson et al., 2002*). Several studies have revealed that antithyroid autoantibodies and thyroid dysfunction may be triggered by IFN-alpha-based therapies for patients with CHC even without pre-existing thyroid abnormalities (*Jadali, 2013*; *Marazuela et al., 1996*). A nationwide population-based cohort study from Taiwan reported that the cumulative incidence of rheumatic diseases was lowest in HCV-uninfected cohorts (95% confidence interval (CI) [8.416%–10.734%]) compared with HCV-infected cohorts treated with (95% CI [12.417%–17.704%]) and untreated HCV-infected cohorts (95% CI [13.585%–16.479%]) (*Cheng et al., 2021*). This finding indicates that HCV infection was associated with some rheumatic diseases.

ANA is widely used for diagnosing several systemic autoimmune diseases, providing not only negative or positive results but also fluorescence patterns. In the past, we detected "antinuclear" antibodies and defined their patterns using immunoassay, including homogeneous, speckled, nucleolar, and nuclear rim (*White & Robbins, 1987*). IIFA Hep-2 allows the identification of autoantibodies targeted to antigens localized in the nucleus and

**Table 4  Clinical feature with higher ANA level over than 1:640.**

| Baseline characteristics | ANA 1:640 (+) | ANA(-) | P-value |
|---|---|---|---|
| Case number | 24 | 234 | |
| Age, years (Mean ± SD) | 69.4 ± 11.8 | 62.3 ± 13.2 | 0.011 |
| Gender Male (*n*, %) | 6(25.0%) | 117(50.0%) | 0.020 |
| Body Mass Index (Mean ± SD) | 24.1 ± 4.1 | 24.8 ± 4.3 | 0.464 |
| Comorbidities | | | |
| Hypertension (*n*, %) | 10(41.7%) | 90(38.5%) | 0.759 |
| Diabetes mellitus (*n*, %) | 3(12.5%) | 48(20.5%) | 0.431 |
| Chronic kidney disease (*n*, %) | 4(16.7%) | 11(4.7%) | 0.039 |
| Cirrhosis | 4(16.7%) | 36(15.4%) | 0.773 |
| HCC (*n*, %) | 0(0.0%) | 9(3.8%) | 1.000 |
| Malignancy (*n*, %) | 3 (12.5%) | 28(12.0%) | 1.000 |
| HCV RNA(log10 IU/ML) (Median(Q1-Q3)) | 5.7(5.19-6.34) | 5.95(4.89-6.46) | 0.573 |
| HCV Genotype (*n*, %) | | | 0.713 |
| 1 | 0(0.0%) | 8(3.4%) | |
| 1a | 0(0.0%) | 19(8.1%) | |
| 1b | 14(58.3%) | 97(41.5%) | |
| 2(2,2a,2b) | 9(37.5%) | 80(34.2%) | |
| 3 | 0(0.0%) | 8(3.4%) | |
| 6 | 0(0.0%) | 8(3.4%) | |
| Mixed | 0(0.0%) | 1(0.4%) | |
| Unclassified | 1(4.2%) | 13(5.6%) | |
| Baseline GPT (U/L) (Median(Q1-Q3)) | 34.5(16.5–72.5) | 44(27–92) | 0.098 |
| Baseline Bilirubin (mg/dL) (Median(Q1-Q3)) | 0.5(0.4–0.8) | 0.6(0.5–0.8) | 0.129 |
| Baseline eGFR (mL/min/1.73m2) (Median(Q1-Q3)) | 77.76(39.21–94.84) | 87.94(68.29–104.38) | 0.044 |
| Fibrosis (*n*, %) | | | 0.652 |
| F0 | 5(29.4%) | 69(40.8%) | |
| F1 | 3(17.6%) | 22(13.0%) | |
| F2 | 3(17.6%) | 29(17.2%) | |
| F3 | 1(5.9%) | 20(11.8%) | |
| F4 | 5(29.4%) | 29(17.2%) | |

cytoplasm and mitotic cells (*Chan et al., 2016*). The ICAP developed a classification tree for most staining patterns. There are homogeneous, speckled, dense, fine speckled, centromere, discrete nuclear dots, and nucleolar reported in nuclear patterns, and cytoplasmic patterns are fibrillar, speckled, reticular/mitochondrion-like, polar/Golgi-like, and rods and rings (*Chan et al., 2022*; *Damoiseaux et al., 2019*). These patterns were correlated with different rheumatological diseases and have been reported to play a significant role as an anti-marker of rheumatic pathology, particularly the AC-2 pattern (*Pashnina et al., 2021*).

**Table 5** Multivariable analysis of factors associated with ANA1:640(+).

| Risk factor | crude OR | *P*-value | Multivariate | |
| --- | --- | --- | --- | --- |
| | | | Adjusted OR | *P*-value |
| Age,yr | 1.05(1.01,1.08) | 0.013 | – | – |
| Gender(Male) | 0.33(0.13,0.87) | 0.025 | 0.31(0.12,0.82) | 0.018 |
| BMI,kg/m$^2$ | 0.96(0.87,1.07) | 0.463 | – | – |
| Hypertension | 1.14(0.49,2.68) | 0.759 | – | – |
| Diabetes mellitus | 0.55(0.16,1.93) | 0.354 | – | – |
| Chronic kidney disease | 4.05(1.18,13.91) | 0.026 | – | – |
| Cirrhosis | 1.1(0.36,3.41) | 0.869 | – | – |
| HCC | | | – | – |
| Malignancy | 1.05(0.29,3.75) | 0.939 | – | – |
| HCV RNA(log10 IU/ML) | 0.9(0.65,1.25) | 0.531 | – | – |
| Baseline GPT (U/L) | 1(0.99,1) | 0.320 | – | – |
| Baseline Bilirubin (mg/dL) | 0.5(0.11,2.35) | 0.383 | – | – |
| Baseline eGFR (mL/min/1.73 m$^2$) | 0.98(0.97,1) | 0.011 | 0.98(0.97,1) | 0.008 |
| Fibrosis | | | – | – |
| F0 | 1 | 0.665 | – | – |
| F1 | 1.88(0.42,8.52) | 0.412 | – | – |
| F2 | 1.43(0.32,6.37) | 0.641 | – | – |
| F3 | 0.69(0.08,6.25) | 0.741 | – | – |
| F4 | 2.38(0.64,8.85) | 0.196 | – | – |

**Notes.**

Multivariate logistic regression analysis was performed to identify factors associated with ANA1:640(+). Selected variables with *p*-value < 0.10 from crude mode (including Age, Gender(Male), Baseline eGFR) for backward elimination procedure in the multivariable model.

There are several diseases; however, autoimmune diseases, such as hyperthyroidism, gastroparesis, and non-alcoholic fatty liver disease, were found to be associated with ANA positivity (*Mitra & Ray, 2020*; *Parkman et al., 2022*; *Xun, 2021*). Although the correlation remains unclear. A study from China on the relationship between ANA and various liver diseases showed that the incidence of ANA positivity in patients with hepatitis B virus (HBV) and HCV infection was 19.1% and 3.9%, respectively. The most common ANA patterns in patients with HBV infection were AC-1, AC-4, AC-21, AC-5, and AC-2. Nevertheless, the most common ANA patterns in patients with HCV were AC-21, AC-4, AC-1, and AC-5 (*Wei et al., 2020*). A recent study analyzed the prevalence and outcomes of autoantibodies in HCV-infected patients. The most common ANA patterns were homogeneous and speckled patterns, accounting for 43% and 36%, respectively, followed by nucleolar (7%), centromeric (7%), and rods and rings (7%) (*Romano et al., 2022*). Furthermore, another study revealed elevated serum markers of autoimmune hepatitis, namely, ANA, immunoglobulin G, smooth muscle actin, anti-LKM1, and soluble liver antigen, in patients with CHC (*Simoes et al., 2019*). However, the authors did not examine the association of these serum markers with the virological features of their patients with HCV infection or did not assess the association of these markers with different ANA patterns.

To the best of our knowledge, this is the first ANA analysis among patients with CHC with viremia according to the ICAP. Furthermore, this study provides insight onto HCV infection and autoimmunity.

In our analysis, the most detected pattern was AC-1, accounting for 25.9%, followed by AC-4, accounting for 25.2%; AC-21, accounting for 9.6%; and AC-8, accounting for 8.9%. In previous reports, the AC-1 pattern was associated with systemic lupus erythematosus, chronic autoimmune hepatitis, and juvenile idiopathic arthritis (*Aringer et al., 2019*; *Petri et al., 2012*). The AC-4 pattern is clinically indicated in systemic autoimmune rheumatic diseases, particularly SjS, lupus erythematosus, dermatomyositis, and systemic sclerosis (*Aringer et al., 2019*; *Betteridge & McHugh, 2016*; *Shiboski et al., 2017*; *Trallero-Araguás et al., 2012*). The AC-21 pattern was found in patients with primary biliary cholangitis and systemic sclerosis (*Shuai et al., 2017*; *Zheng et al., 2017*). Finally, the AC-8 pattern was found in patients with systemic sclerosis (*Ceribelli et al., 2010*; *Mahler, Fritzler & Satoh, 2015*). Our patient population has no clinical manifestations of the aforementioned rheumatological conditions. There is no prior research on ANA patterns in CHC with viremia, and we provide additional information regarding the new ANA pattern among patients with HCV infection. Thus, whether HCV per se causes the presence of such autoantibodies or triggers the subsequent development of the aforementioned rheumatological conditions requires further Investigation.

ANA positivity was mostly found in females, accounting for 63.5%. This finding is consistent with the findings of a study on the relationship between sex and autoimmune diseases, which reported that females are more likely to be ANA-positive than males (*Whitacre, 2001*). A study revealed that ANA-positive patients with CHC were significantly associated with higher levels of aspartate aminotransferase, alkaline phosphatase, and alpha-fetoprotein (*Peng et al., 2001*). Another study reported a faster rate of liver fibrosis in ANA-positive patients with CHC but without statistical significance (odds ratio = 1.8; $p = 0.1452$) (*Yee et al., 2004*). However, a recent study showed no correlation between cirrhosis, hepatic decompensation, HCC, and survival during the 10-year follow up in patients with CHC with or without autoantibody positivity (including ANA, ASMA, AMA, and anti-LKM (*Gilman et al., 2018*). Our data showed that the ANA(+) group had a lower HCV RNA viral load and higher prevalence of advanced fibrosis (F3 and F4). Additionally, in this study, we noticed higher ANA titers after adjusting for age and sex, which showed a significant association with a lower eGFR ($p = 0.008$). This study has some limitations. First, this is a retrospective study in a single institution involving a small sample. However, our study has one strength in evaluating hepatitis C viremic patients with their ANA profiles. Some patients had a positive or higher ANA titer with HCV viremia, as hepatitis C infection is associated with some autoimmune diseases. Knowing the baseline ANA profiles paves the way for further studies comparison of the evolution of ANA profile change following hepatitis C eradication. Second, there was no control group for comparison, and we didn't have data on other autoantibodies, including ASMA, AMA, and anti-LKM1, in our cohort for further analysis. Third, we only analyzed the patients' pretreatment data. A long-term follow-up study on patient outcomes, including fibrosis stage, hepatic decompensation,

cirrhosis, HCC, and other comorbidities, is required to assess ANA pattern changes after HCV eradication.

## CONCLUSIONS

We found that the overall prevalence of ANA positivity in patients with CHC was 28.7%. The most detected ANA patterns were AC-1, AC-4, AC-8, and AC-21. ANA-positive patients revealed a lower HCV RNA viral load and a higher prevalence rate of advanced fibrosis, although there was no statistical significance. Furthermore, higher ANA titer values (>1:640) were significantly associated with a lower eGFR.

### Funding

This research was funded by the Changhua Christian Hospital (110-CCH-IRP-040). The funders had no role in study design, data collection and analysis, decision to publish, or preparation of the manuscript.

### Grant Disclosures

The following grant information was disclosed by the authors:
The Changhua Christian Hospital: 110-CCH-IRP-040.

### Competing Interests

The authors declare there are no competing interests.

### Author Contributions

- Shun-Wen Hsiao conceived and designed the experiments, authored or reviewed drafts of the article, and approved the final draft.
- Chuan-San Fan performed the experiments, prepared figures and/or tables, and approved the final draft.
- Hsu-Heng Yen conceived and designed the experiments, authored or reviewed drafts of the article, and approved the final draft.
- Siou-Ping Huang analyzed the data, prepared figures and/or tables, and approved the final draft.
- Yang-Yuan Chen performed the experiments, authored or reviewed drafts of the article, and approved the final draft.
- Pei-Yuan Su conceived and designed the experiments, authored or reviewed drafts of the article, and approved the final draft.

### Human Ethics

The following information was supplied relating to ethical approvals (i.e., approving body and any reference numbers):

The Ethics Committee of Changhua Christian Hospital approved the study protocol used in this study(CCH IRB No: 220327).

## Data Availability

The raw data is available in the Supplemental Files.

## Supplemental Information

Supplemental information for this article can be found online at http://dx.doi.org/10.7717/peerj.14200#supplemental-information.

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
