# Peer review of "A retrospective study of prevalence and pattern of international consensus on ANA patterns among patients with hepatitis C virus infection"

_PeerJ, doi:10.7717/peerj.14200_

## Round 0.1 · original submission · Major Revisions

Your manuscript 'A study of prevalence and pattern of international consensus on ANA patterns among patients with hepatitis C virus infection' has been assessed by our reviewers. Based on these reports, and my own assessment as Editor, it needs major revision. Once you have carried out the essential revisions suggested by our reviewers, send it back.

Reviewer 1 ·

Basic reporting

The entire manuscript need to be checked for language and grammar.

Introduction: The Introduction section can be improved if the knowledge gap is highlighted. Please comment on what is already known and how this piece of research can bridge the knowledge gap. It is mentioned in line 24-27 that autoimmune diseases and auto antibodies are triggered by IFN therapy. However, the population under study for this research were analysed for auto antibodies before DAA treatment (line 32). Kindly clarify the importance of measuring ANA only before start of DAA therapy. A better plan of study would have been to compare the ANA status with two modes of therapy. ANA is a non-specific parameter and positivity is seen in many conditions, hence association with patient characteristics may not be much helpful. The English language can be improved to make the text understandable, Examples- Line 4,5,24,25,26,33.

Raw data has been shared. The full form of abbreviations at the top of column in excel sheet must be shared.

Experimental design

Materials & Methods: Please provide whether the hepatitis C virus infection was acute or chronic. In line 47, it is mentioned that the dilutions started from 1:10, 1:100 and 1:1000. Please justify why 1:10 and 1:1000 dilutions were used? If three dilutions were used , how to justify the statement in line 55-‘The ANA was resumed positive as titer larger than 1:320. I assume 1:320 was not used. The grammar from lines 45-55 needs to be checked and rewritten.

Validity of the findings

Results: Line 101- mentions of more than 1 pattern detection. Please specify how this was detected and what criteria were taken into consideration for such positivity. Table 4 takes into consideration of 1:640 titer positivity and compares with clinical features. Please justify the reason behind such comparisons. In all tables IQR is mentioned, however, it is depicted as range and not IQR.

Discussion: Lines 109-115 is a repeat description of results. Discussion is also a repetition of information in Introduction; Lines 118-120, 122-126, 134-142. I feel the discussion does not reveal any new information.

Additional comments

The importance/rationale of the study need to be highlighted. The discussion can focus on new findings of the study as well the impact it will have on clinical practice in the near future. Since the data included very limited number of HCV who were positive with ANA, data from multiple centers can be collected to increase the sample size. The other suggestion is to increase the time span of data collection period to increase the sample size. Another suggestion that can be considered, since data is collected from medical record the clinical and laboratory data can be collected after DAA therapy. The English language and grammar needs to be checked for the entire manuscript.

Ethics approval statement in the manuscript is appropriate and it meets the requirements of the journal. There is no identifiable information in the file. The data has been obtained from the medical records of patients and obtained ethically. However, the protocol title in the Clinical Trials Approval certificate does not match with the submitted proposal sent for review.

Reviewer 2 ·

Basic reporting

The authors analyzed the prevalence and pattern of ANA among patients with HCV infection using the International Consensus on ANA Patterns. It is novel, but the significance of the present study is unclear, or not well written. Most references are timeworn, please add some new literature.

Experimental design

No.

Validity of the findings

1. It is not enough just to analyze the correlation between ANA and eGFR. Please add the virological response to DAA or IFN-a. Is there any difference in virological response or side effects between patients with ANA (+) and (-)? The outcomes or prognosis may be more interesting for researchers.
2. Patients in ANA positive group (and ANA 1:640) were older, and more patients had chronic kidney disease, that may be the reason for the lower eGFR, so a multivariate logistic analysis may be useful to find whether ANA (+) was an independent risk factor for impaired eGFR.
3. There have been several reports about prevalence and pattern of ANA in patients with CHC, please discuss the difference between the present new pattern model and previously reported pattern models. In my opinion, it is just the significance of the present study.

Additional comments

1. It could not conclude that ALT was lower in ANA (+) group, because the P value was 0.574 (line 95).

---

## Round 0.2 · Minor Revisions

Your manuscript 'A study of prevalence and pattern of international consensus on ANA patterns among patients with hepatitis C virus infection' has been assessed by our reviewers. Based on these reports, and my own assessment as Editor, it needs minor revision. Once you have carried out the essential revisions suggested by our reviewers, send it back.

Reviewer 1 ·

Basic reporting

The basic reporting and English language has improved. Recent literature has been incorporated.

Experimental design

Knowledge gap explained. A detailed methodology would be good. Mention the various dilutions that were used for the analysis. Since ANA pattern identification and positivity is subjective in nature, how was the various patterns identified, How was the bias reduced?
Total number of ANA positive patients were 74, and each pattern had also very few number of each case. Therefore a percentage as depicted in Table2 is not a correct representation, it is better to give it as a ratio( no of positive / total positive samples). Table 3 depicts mixed patterns, please describe how the two mixed patterns were identified as two or more patterns can overlap and affect their identification. In table 3, do not present as percentages.

Validity of the findings

Raw data provided. The end of discussion must mention how this knowledge derived from this research would be helpful in clinical management of patients.

Reviewer 2 ·

Basic reporting

It is revised well, and I have no question.

Experimental design

It is revised well, and I have no question.

Validity of the findings

It is revised well, and I have no question.

---

## Round 0.3 · accepted · Accept

I am writing to inform you that your manuscript - A retrospective study of prevalence and pattern of international consensus on ANA patterns among patients with hepatitis C virus infection - has been Accepted for publication.